# Influence of DNA Methylation on Vascular Smooth Muscle Cell Phenotypic Switching

**DOI:** 10.3390/ijms25063136

**Published:** 2024-03-08

**Authors:** Chanthong Yorn, Hyunjung Kim, Kyuho Jeong

**Affiliations:** Department of Biochemistry, College of Medicine, Dongguk University, Gyeongju 38066, Republic of Korea; yornchanthong168@gmail.com (C.Y.); hunjung612@naver.com (H.K.)

**Keywords:** vascular smooth muscle cells, DNA methylation, vascular diseases, DNMTs

## Abstract

Vascular smooth muscle cells (VSMCs) are crucial components of the arterial wall, controlling blood flow and pressure by contracting and relaxing the artery walls. VSMCs can switch from a contractile to a synthetic state, leading to increased proliferation and migratory potential. Epigenetic pathways, including DNA methylation, play a crucial role in regulating VSMC differentiation and phenotypic flexibility. DNA methylation involves attaching a methyl group to the 5’ carbon of a cytosine base, which regulates gene expression by interacting with transcription factors. Understanding the key factors influencing VSMC plasticity may help to identify new target molecules for the development of innovative drugs to treat various vascular diseases. This review focuses on DNA methylation pathways in VSMCs, summarizing mechanisms involved in controlling vascular remodeling, which can significantly enhance our understanding of related mechanisms and provide promising therapeutic approaches for complex and multifactorial diseases.

## 1. Introduction

Vascular smooth muscle cells (VSMCs) play a significant role within the linings of blood vessels, mainly in the medial layer, where they regulate blood flow, vessel tone, and blood pressure [1,2]. They have remarkable plasticity and respond to various environmental signals [3]. In the arteries of mature individuals, VSMCs are surrounded by an underlying membrane composed of collagen type IV, laminin, and heparan sulfate proteoglycan [4,5]. Differentiated VSMCs have low proliferation rates, exhibit little synthetic activity, and have contractile proteins and ion channels crucial for their contractility [6,7]. However, VSMCs undergo phenotypic switching after arterial injury, becoming osteochondrocyte-like cells, foam cells, or myofibroblasts [8,9]. This process includes a decrease in the expression of genes related to VSMC differentiation and an increase in VSMC migration, proliferation, and the production of extracellular matrix (ECM) elements necessary for vascular repair [10,11], which is critical for the formation of arterial lesions and vascular remodeling [6]. Phenotypically modified VSMCs contribute to various cardiovascular diseases such as atherosclerosis, aortic aneurysm, transplant vasculopathy, intimal hyperplasia and restenosis, vascular calcification, hypertension, and aberrant tumor vasculature [12,13,14,15]. VSMCs differ from musculoskeletal and cardiovascular myocytes in that they exhibit significant cellular plasticity, and their phenotype is significant in response to external stimuli and signals. VSMCs are often seen in quiescent contractile conditions with a limited number of protein synthesis organelles, such as rough endoplasmic reticulum (ER) and free ribosomes, and rarely proliferate in mature functional vessels [16]. They produce proteins responsible for controlling processes related to smooth muscle-specific contractions and contractile functions, including channel proteins, SM-myosin heavy chain (SM MHC, *Myh11*), Smooth muscle alpha-actin (SMA, *Acta2*), SM22 (*Tagln*), h-caldesmon (*Cald1*), calponin (*Cnn1*), leiomodin, and smoothelin (*Smtn*) [17,18,19,20]. VSMC phenotypic alteration may be monitored by a decrease in myofilament density, contractile marker genes, and contractility [21].

## 2. Vascular Smooth Muscle Cell Phenotype Switching in Disease

VSMCs in blood arteries can undergo phenotypic change from a contractile to a synthetic or proliferative state, which leads to abnormal vascular remodeling and vascular lesions in various vascular diseases [22]. VSMCs undergo alterations in their characteristics as a result of processes such as vascular development, healing of vascular injuries, and the abnormal restructuring of blood vessels in individuals affected by cardiovascular disorders [23]. VSMC phenotypic switching is associated with several disorders, including the following:

**Atherosclerosis** is a complex condition that impacts various cell categories, including lymphocytes, macrophages, endothelial cells (ECs), and VSMCs. The primary factor for coronary artery disease and a chronic condition of the main blood vessels [24], leading to myocardial infarction, stroke, heart failure, and peripheral arterial disease (Figure 1) [25]. It has been hypothesized that phenotypically altered VSMCs are essential for the development of atherosclerosis. In Western society, it accounts for almost 40% of all fatalities [26]. Interestingly, VSMC differentiation markers are significantly reduced while intimal VSMCs within eccentric atherosclerotic lesions invade the lumen, and median VSMCs are essential for adaptive outward remodeling, which helps maintain blood flow [27]. When inflammatory cells release chemokines and growth factors, such as foam cells made from macrophages, VSMCs from the vascular media move toward the intima and undergo proliferation, contributing to the formation of atherosclerotic plaque. According to findings from research conducted by Misra, disturbances in the integrin-β3 pathway were observed to lead to the polyclonal development of atherosclerotic lesions. This indicates that a substantial number of medial VSMCs maintain their ability to proliferate, and external signals may play a crucial role in regulating the proliferation and clonal expansion of VSMCs [28]. Furthermore, Single-cell RNA sequencing (scRNA-seq) studies have identified various clusters of VSMCs within the media. While it remains unclear if a specific VSMC cluster can clonally expand, evidence suggests that VSMCs within atherosclerotic lesions originate from a limited number of clones. These VSMC clones can give rise to various lesion cell types, including fibrous cap cells, macrophage-like cells, mesenchymal stem cell-like cells, and osteochondrogenic cells [29,30,31,32,33,34,35,36]. It is hypothesized that transitioning to a mesenchymal stem cell-like state may precede and be necessary for transitioning to other lesion phenotypes. Additionally, VSMC-derived lesion cells may originate from the medial Sca1+ VSMC population. Phenotypic modulation also occurs within the media during atherosclerosis, independent of clonal expansion [37,38]. The process of transitioning from a specialized or contracting form in the middle layer of a structure to a de-differentiated or creative state in the innermost layer is referred to as "phenotypic modulation” or “switch”. This transition occurs along a spectrum between these states [39]. This process involves enhanced proliferative activity, the creation of numerous ECM components and proteases, and the downregulation of contractile proteins. Defining the level of plasticity in phenotypically modified VSMCs and identifying molecular pathways that govern important changes in VSMC phenotype after vascular injury are major challenges in the field [7,40]. 

**Pulmonary arterial hypertension (PAH)** is a progressive condition characterized by a rise in resistance within the pulmonary blood vessels. If not addressed, this can ultimately result in the failure of the right ventricle and can often be lethal [41,42]. The Sixth World Symposium for Pulmonary Hypertension defined PAH as a class of cardiovascular conditions with a mean pulmonary artery pressure of at least 20 mm Hg, a pulmonary artery occlusion pressure of at least 15 mm Hg, and a pulmonary vascular resistance of at least three Woods units (one Woods unit equals 80 dynes per second per cm^−5^) [43,44]. The etiology of PAH is complicated and attributed to a combination of genetic vulnerability and adverse events [45]. Recent studies have found that DNA methylation is a distinctive feature of pulmonary artery smooth muscle cells in patients with pulmonary hypertension compared to those in healthy individuals [46,47]. Various variants have been identified as potential causative factors in PAH, including KCNK3, GDF2, EIF2AK4, BMPR2, TBX4, SOX17, CAV1, ATP13A3, SMAD9, AQP1, ACVRL1, and ENG, through exome sequencing and patient cohort surveys [48,49,50,51,52,53,54,55,56,57,58]. The changes in the blood vessel structure associated with pulmonary arterial hypertension (PAH) are characterized by a shift in the behavior of VSMCs from their essential “resting” state to more active “pro-proliferative,” “resistant to cell death” and “promoting inflammation” states. This transformation leads to the accumulation of ECM components, mainly collagen, and an increase in the thickness of pulmonary arterioles. In the initial stages of PAH, pathological VSMCs de-differentiate by reducing MYH11 and increasing PDGFR-b, leading to clonal proliferation and distal migration [59].

**Intimal hyperplasia (IH)**, a form of vascular remodeling, occurs when VSMCs proliferate and migrate into the intima. This process is observed in several pathological conditions, including atherosclerosis and pulmonary arterial hypertension, as well as during restenosis following procedures like angioplasty and vein grafting [60,61]. In normally functioning and developed blood vessels, the inner layer is primarily composed of endothelial cells and contains a small number of contractile VSMCs. When there is physical damage to the blood vessel, like during procedures such as arterial balloon angioplasty or stenting, the VSMCs lose their specialized characteristics through a process called de-differentiation. In injury-induced situations, local inflammatory cues encourage VSMC phenotypic switching to a “synthetic” phenotype and create a new layer, commonly referred to as the neointima, between the lumen and the internal elastic lamina. These intimal VSMCs produce and secrete many ECM elements and remodeling-related proteins, such as matrix metalloproteinases [61,62]. VSMC proliferation, stimulated by cytokines and growth factors in response to vascular damage, may be responsible for up to 90% of total intima proliferation (Figure 2) [61]. 

Xu et al., characterized VSMC transcriptomic phenotypes in neointimal hyperplasia using scRNA-seq, identifying macrophage-like VSMCs expressing high levels of C3 and Cd74 as a key population. TNF-α-induced activation of Sox10 via PI3K/AKT signaling promotes VSMC transdifferentiation and vascular inflammation [63]. Another research group employed scRNA-seq to investigate the mechanisms of in-stent restenosis. They identified transitional cell types between VSMCs and fibroblasts in both normal and stenotic arteries and classified VSMCs into six distinct clusters based on significant phenotypic changes within stenotic arteries. Moreover, N-myristoyltransferase 1 (NMT1) was verified as a credible VSMC synthetic phenotype marker. Disease susceptibility gene analysis confirmed the involvement of classical genes associated with in-stent restenosis and proposed that novel target genes like Cyp7a1 and Cdk4 be used for future validation studies [64]. 

Despite current therapeutic techniques, such as drug-eluting stents targeting VSMC proliferation, there remain concerns regarding in-stent stenosis and delayed re-endothelialization, especially in high-risk patients. These obstacles are particularly challenging in the treatment of peripheral arterial disease [65,66]. 

**Aortic aneurysm (AA)** is a common and severe dilated arterial condition related to thoracic aortic aneurysm and abdominal aortic aneurysm. AA is associated with a high mortality rate [67]. A defining characteristic of AA involves the enhancement and diminishment of the structural integrity of the vessel wall, which may lead to either tearing or bursting. This condition stands as a prominent contributor to mortality in developed countries, accounting for a substantial fatality rate of approximately 90% in instances of rupture [68,69]. Risk factors for AA include male gender, older age, smoking, atherosclerosis, dyslipidemia, hypertension, obesity, and a family history of the disease [70,71]. AA is linked to chronic inflammation, VSMC remodeling, apoptosis, and ECM degradation [72]. VSMCs are critical to maintaining the strength of the healthy vasculature and adapting to external stimuli. However, VSMC dysfunctions, such as apoptosis and phenotypic switching, have been shown to contribute to AA development [73,74]. Mutations in various genes linked to a differentiated contractile phenotype (ACTA2, MYH11, PRKG1, MYLK, and TGF-β signaling) have been associated with human AA [75]. In healthy arteries, mechanical stresses and contraction maintain the differentiated contractile phenotype, while mechanotransduction gene dysfunction can result in aneurysm development [76]. If the mechanism that triggers and inhibits VSMC phenotypic switching during AA development resulting in vascular impairment can be understood, novel preventive and therapeutic approaches may be developed for AA.

**Chronic kidney disease (CKD)** is characterized by a gradual decline in kidney function. The link between chronic kidney disease (CKD) and cardiovascular death is significantly impacted by changes in blood vessels, particularly atherosclerosis and vascular calcification in the inner and middle layers [77,78,79]. In CKD, factors like inflammatory cytokines, uremic toxins, hypercalcemia, and hyperphosphatemia trigger calcification, mainly in the middle layer of blood vessels [80,81,82]. Vascular calcification involves various processes in VSMCs, including apoptosis, the transformation into osteochondrogenic cells, extracellular vesicle release, excessive calcium accumulation, and cellular aging [83].

Persistent inflammation in CKD leads to the accumulation of reactive oxygen species (ROS) and inflammatory proteins within blood vessels, activating factors like runt-related transcription factor 2 (RUNX2) and BMP2, which promote the expression of osteocytic VSMCs within vessel walls, contributing to mineralized matrix formation [84,85,86]. During VSMC transdifferentiation, there is a shift from VSMC indicators like SM22α to osteochondrogenic markers such as RUNX2, alkaline phosphatase, osteocalcin, and osteopontin [62,67,87,88]. Normally, the activities of RUNX2 and BMP2 are balanced by inhibitors produced in VSMCs like matrix Gla protein (MGP). However, in CKD, these inhibitors are often suppressed, leading to increased calcification in the vascular media. Medial calcification results from the breakdown of the elastin-rich matrix, initiated by extracellular vesicles containing calcium phosphate crystals released by VSMCs [89,90,91,92,93]. 

In CKD, VSMC apoptosis and autophagy also play significant roles in vascular calcification. CKD-induced oxidative and uremic stress increase VSMC apoptosis, leading to the release of apoptotic bodies rich in calcium. These bodies, along with extracellular DNA, accumulate on the vascular media’s ECM, inducing calcification. Elevated mineral levels and CKD-related stress further exacerbate this process, leading to the loss of contractile phenotype proteins and subsequent vasculature calcification [94,95,96,97]. Vitamin D supplementation may help to limit calcification by regulating RUNX2 expression, while magnesium supplementation could hinder VSMC transdifferentiation. Deficiency in osteopontin (OPN) also contributes to calcification, suggesting its role in mineralization regulation. Further clinical investigation is warranted to explore these potential therapeutic avenues [98,99,100].

**Diabetes mellitus** is a complex condition characterized by elevated blood glucose levels, which can lead to organ damage, particularly affecting the kidneys [101]. The impact of glucose metabolism on VSMCs is significant, especially in the media layer [102]. Diabetic patients often exhibit elevated glucose levels, leading to the upregulation of F-actin, a-SMA, and cytoskeletal components. Type 2 diabetes (T2D) is associated with chronic inflammation, characterized by the production of pro-inflammatory factors like TNF-a, IL-a, FGF21, and PDGF at sites of high glucose exposure. This inflammatory response involves pathways such as PI3K/Akt and NF-kB, contributing to VSMC de-differentiation, proliferation, and migration, which can trigger the additional release of pro-inflammatory agents by the VSMCs themselves, worsening vascular dysfunction [67,103,104,105,106,107,108,109]. Additionally, VSMCs in individuals with T2D exhibit increased expression of the antiapoptotic protein Bcl-2, leading to resistance to cell death, enhanced proliferation, and a thicker media layer [110]. Prolonged exposure to elevated glucose levels induces oxidative stress in vascular cells, a crucial factor in the progression of atherosclerosis [111,112,113,114].

Individuals with diabetes often experience severe vascular calcification, primarily involving VSMCs. Elevated glucose levels exacerbate atherosclerosis and vascular calcification by amplifying the effects of oxidized low-density lipoprotein. Under oxidative stress conditions and high glucose levels, VSMCs transform into cells resembling osteoblasts, releasing bone-related proteins. Increased leptin levels also activate the BMP osteogenic signaling pathway, facilitating VSMC transformation into osteoblasts and promoting vascular calcification [115,116,117]. Dyslipidemia, particularly acetylated low-density lipoprotein, enhances VSMC osteogenic characteristics, while high-density lipoprotein suppresses this process [118,119]. Elevated cholesterol levels increase oxidative stress and expedite vascular calcification triggered by vitamin D [120]. Triglyceride levels are an independent predictor of coronary arterial calcification progression in individuals with diabetes. Additionally, some specific research demonstrates an association of hyperlipidemia with Wnt/β-catenin signaling, which is crucial in vascular calcification [121,122]. Insulin stimulates the vascular endothelium in order to release nitric oxide, which reduces intimal calcification and inhibits VSMC formation [123]. However, insulin resistance increases the influx of free fatty acids into the liver, leading to the elevated hepatic uptake of triglycerides and the synthesis of very low-density lipoprotein, thereby increasing the risk of vascular calcification [124].

**The generalized arterial calcification of infancy (GACI)** is characterized by the calcification of major arteries, including large and medium-sized vessels, accompanied by intimal proliferation, affecting the coronary arteries in particular [125]. Patients with GACI show decreased levels of inorganic pyrophosphate (PPi) in their plasma and diminished activity of the ectonucleotide pyrophosphatase/phosphodiesterase 1 (ENPP1) enzyme. Mutations in the ENPP1 gene, leading to ENPP1 enzyme deficiency, are the primary causative factors of this, with contributions from variations of ATP-binding cassette sub-family C member 6 (ABCC6) [126,127]. Calcification initiation involves calcium deposition in the elastic lamina, extending into the intima and media [128]. VSMCs play a crucial role in ectopic arterial calcification, generating matrix vesicles and exosomes within the calcified region and initiating mineralization and bone matrix development within the vessel wall [8]. ENPP1 deficiency leads to reduced PPi levels, increased pro-proliferative extracellular ATP levels, and decreased anti-proliferative AMP and adenosine levels, accelerating VSMC proliferation and arterial stenosis [129]. ENPP1 converts extracellular ATP into AMP, resulting in PPi generation. This inhibits hydroxyapatite formation, regulating chondrogenesis in order to prevent soft tissue calcification. Recombinant human ENPP1-Fc protein administration inhibits VSMC proliferation, showing efficacy in preventing aortic calcification and myocardial infarction in mouse models with ENPP1 deficiency. ENPP1 enzyme replacement is being explored as a potential therapeutic strategy for addressing intimal hyperplasia in GACI [130].

**Pseudoxanthoma elasticum (PXE)** is a genetic disorder resulting from mutations in the ABCC6 gene, causing abnormal mineralization in elastin-containing tissues, particularly in medium-sized arteries [131,132,133]. This leads to increased arterial stiffness and reduced compressibility in PXE patients [134]. VSMCs, linked to a porous arrangement of elastin and collagen strands within the arterial media, have a crucial role in the formation of vascular calcifications observed in PXE and various other conditions [135]. In typical circumstances, VSMCs exhibit natural protection against calcification through substances like PPi and MGP [136]. However, under abnormal conditions, VSMCs experience programmed cell death, aging, and increased oxidative stress, leading to their transformation into cells resembling osteoblasts. This transformation contributes to phosphate-induced vascular calcification, resulting in two main outcomes for VSMCs: a shift towards a bone-forming phenotype and apoptosis-dependent mineralization [137,138,139]. Studies suggest that hydroxyapatite deposits can stimulate VSMCs to adopt a bone-forming phenotype, indicating a compensatory response to ectopic calcification [140,141,142]. Significantly, VSMCs interact with hydroxyapatite; they exhibit decreased SM22α expression and increased BMP2 expression, with effects being dependent on dosage [143]. Recent findings suggest that arterial calcification in PXE is primarily influenced by age rather than levels of circulating PPi, indicating limitations in using PPi as a marker of disease severity [144].

Researchers have explored arterial remodeling in PXE [145], highlighting the crucial role of VSMCs within the vessel wall’s medial layer. These VSMCs, connected to elastin and collagen fibers, maintain vessel elasticity to accommodate heartbeat volume and regulate pressure pulsations. Their activity affects ECM stress, thus influencing vessel diameter and stiffness [83,146]. The absence of magnesium ions is associated with various cardiovascular issues in PXE patients, including high blood pressure, arterial hardening, impaired blood vessel function, and arterial calcification [147]. Supplementing magnesium results in reduced carotid arterial thickness in ABCC6−/− mice [148,149].

## 3. DNA Methylation and Its Mechanism in VSMC Phenotype Switching

Epigenetics involves changes in gene expression control without altering the DNA sequence itself. These changes are inherited through modifications of the genetic code’s structure. These modifications include DNA methylation, histone modification, and the involvement of non-coding RNAs such as microRNAs [150,151]. Epigenetics is relevant across various fields, including cardiovascular diseases, obesity, neurodegenerative diseases, type 2 diabetes mellitus, inflammation, insulin resistance, and immune disorders [150,152,153,154].

DNA methylation, catalyzed by DNMTs (DNA methyltransferases), predominantly occurs at the 5 positions of cytosine (5-methylcytosine or 5-mC), primarily developing at CpG dinucleotides, but also at non-CpG sites such as CpA, CpC, and CpT. Its main roles involve either silencing or activating genes, depending on the methylated areas [155]. Three DNMTs are capable of methylating cytosine: DNMT1, DNMT3A, and DNMT3B (Figure 3). DNMT1 serves as a key enzyme in preserving DNA methylation during DNA replication. Conversely, DNMT3A and DNMT3B act as de novo DNA methylation enzymes that regulate gene expression [156,157]. DNA methylation can suppress transcriptional activity by recruiting and binding methyl-CpG-binding domain (MBD) proteins like MeCP2 [158,159,160]. CpG methylation at key regulatory regions can contribute to transcript downregulation through changes in genomic DNA folding, modifications in transcription factor accessibility, and an association with histone marks [161].

Recent research has revealed that DNA methylation is a key factor in controlling the process of vascular remodeling, influencing various behaviors such as migration, proliferation, differentiation, and calcification. Diseases associated with vascular aging often exhibit increased VSMC proliferation as a key characteristic [163]. While direct DNA methyltransferases have been identified, DNMTs play a role in hindering gene expression through DNA methylation, thereby impacting VSMC proliferation. Specifically, the hypermethylation of PTEN, ER-α, and MFN2 results in the reduced expression of their mRNA, promoting VSMC proliferation [164,165,166]. On the contrary, the demethylation of the PDGF gene enhances PDGF mRNA and protein expression, thereby stimulating both VSMC proliferation and migration [167]. The hypomethylation of the HIF-1α gene also contributes to VSMC proliferation and migration [168]. Several studies have shown that enzymes from the ten-eleven translocation (TET) family may oxidize 5-mC into 5-hydroxymethyl-cytosine (5-hmC) (Figure 4) [169,170]. Liu et al. reported a significant reduction in TET2 expression within the neointima or coronary atherosclerotic plaques. This reduction resulted in a decrease in 5-hmC enrichment and led to the hypermethylation of MYH11, SRF, and MYOCD, subsequently influencing the results of VSMC differentiation and ultimately contributing to worsened vascular remodeling (Figure 5) [25,171]. Therefore, it is critical to understand the function of DNA methylation alterations in VSMCs and vascular remodeling. Importantly, DNA methylation’s influence on gene expression can endure, even after the elimination of underlying risk factors [172,173]. 5-aza-2’-deoxycytidine (5-aza), a DNMT inhibitor, has the potential to reverse decreased DNA methylation, thus reducing gene expression, and it is now approved for the treatment of myelodysplastic syndrome. 5-aza therapy has been found to reduce neointimal development, which supports the hypothesis that 5-aza slows vascular remodeling via DNA demethylation in VSMCs [174,175].

On the other hand, the ECM has been found to influence the phenotype of VSMCs [176]. A study by Jiang, J.X. et al. revealed that VSMCs exhibited significant DNA methylation changes when plated on native collagen compared to denatured collagen. Plating on damaged collagen increased VSMC proliferation, which was reversed using 5-Aza. Additionally, this study observed an upregulation of nuclear DNMT3 expression in VSMCs on damaged collagen [177]. This study also showed that the matrix caused significant changes in DNA methylation in a subset of CpG sites near genes involved in VSMC differentiation. The matrix has a significant impact on the control of the VSMC phenotype, affecting altered gene expression, DNMT localization, and specific changes in DNA methylation [177,178,179]. The phenotype of VSMCs is also linked to collagen type XV alpha 1 (COL15A1), which has a negative association with cell proliferation and migration. The COL15A1 gene may be regulated by DNA methylation, as evidenced by its responsiveness to treatment with 5-aza and increased protein levels with passage-dependent reductions in DNA methylation [180]. COL15A1 is a non-fibrillar that self-trimers via a core triple-helix domain, bridging vast collagenous bundles in vivo, and is present in the basement membrane of several tissues [181,182]. As a component of the extracellular matrix, COL15A1 influences the proliferative and migratory characteristics of proliferating VSMCs and may play a crucial role in the development of atherosclerosis. The upregulation of COL15A1 expression was observed in an atherosclerotic plaque and confined to the atherosclerotic cap [180,183]. The reduction in COL15A1 expression increased VSMC migration and reduced proliferation. Furthermore, there was a reduction in methylation levels within the COL15A1 gene. This decreased methylation led to increased gene expression, which had the potential to impact the identifiable traits of VSMCs and contribute to the development of atherosclerosis [180]. Focal adhesion kinase (FAK) is a protein tyrosine kinase that plays a critical role in transmitting extracellular signals through integrin or growth factor receptors [184,185]. Our recent study found that FAK activation increases DNA methylation, which promotes VSMC de-differentiation [163]. The FAK-DNMT3A axis controls VSMC phenotypic switching and determines whether VSMCs differentiate. In normally functioning blood vessels, FAK resides within the nucleus of VSMCs, where it plays a role in preserving the specialized condition of these cells. It performs this by disrupting the DNMT3A protein, which in turn helps to limit the extent of DNA methylation on contractile genes related to VSMCs. When arterial damage occurs, FAK translocates to the cytoplasm and becomes activated, leading to increased DNA methylation via DNMT3A. This increase in DNA methylation promotes VSMC de-differentiation and synthetic phenotypes during vascular remodeling [163]. Inducing nuclear FAK localization, using either the pharmacological inhibition of FAK activity or the introduction of genetic FAK kinase-dead mutation, enhances the differentiation of VSMCs by reducing the levels of DNMT3A expression and DNA methylation. This increase in VSMC contractile gene expression results in the promotion of VSMC differentiation. This study also revealed that cytoplasmic FAK activation affects DNA methylation through DNMT3A, resulting in VSMC de-differentiation and the emergence of synthetic phenotypes during vascular remodeling. Nuclear FAK has been demonstrated to function as a scaffold protein, facilitating protein degradation by enlisting a specific target protein and its associated E3 ligases [186]. While DNMT3A and DNMT3B share structural similarities, nuclear FAK selectively interacts with DNMT3A rather than DNMT3B, triggering ubiquitination and subsequent proteasomal degradation through the E3 ligase TRAF6. Furthermore, DNMT3A, but not DNMT3B, is responsible for silencing VSMC contractile genes (Figure 5) [163].

DNA methylation regulates the phenotypic markers of VSMCs, which plays an important role in VSMC phenotype switching [187]. OPN is a versatile phosphoprotein synthesized by various cells, and VSMCs demonstrate elevated OPN levels during their phenotype transition [188,189]. In the walls of healthy veins, the presence of OPN is at a minimum level. However, when triggered by specific cytokines that encourage the growth of VSMCs, the expression of OPN can rise notably [190]. SMA is another protein specific to VSMCs that plays a critical role in cell contraction, which gradually increases in VSMCs with a contractile phenotype [191]. Jiang, H. et al. reported that, in the neointima of varicose veins, VSMCs express OPN and SMA less frequently than normal veins, suggesting that VSMCs change from showing contractile properties to exhibiting a synthetic phenotype in varicose veins [192]. Integrin β3 is a crucial OPN receptor that controls VSMC migration and proliferation. OPN binds to integrin β3 to stimulate VSMC migration and binding, thereby promoting cellular responses [193]. The coexistence of two markers located on the surface of the cell may be connected to VSMC phenotype switching in varicose vein neointima. Modifying promoter regions and other regulatory elements through methylation interferes with the ability of transcription factor complexes to bind to DNA and disrupts the gene translation process [194]. The reduced methylation of promoter areas in the OPN and integrin β3 genes amplify their activity in terms of varicosity. When OPN binds to integrin β3 on the cell membrane of VSMCs, it activates signaling pathways like the integrin β3-FAK pathway. This leads to increased proliferation and migration of VSMCs, causing them to adopt a synthetic phenotype. This change is important for the formation of neointima in varicose veins [192].

DNA methylation can lead to long-term gene silencing [195], but its involvement in controlling gene expression specific to certain cell types is not as well defined, and this might be influenced by the count and methylation condition of CpG sites within a particular promoter area [196,197]. Methylated cytosines can hinder the binding of transcription factors to cis-regulatory elements or engage with proteins containing methyl-binding domains, like MBDs1-4, MeCP2, and Kaiso, that act as repressors of transcription [198,199]. Bartels, S.J. et al. showed that RBPJ (recombination signal binding protein for immunoglobulin kappa J region), a Notch transcription factor, can bind to methylated DNA at the G(Cm/T) GGGA sequence [200]. RBPJ/CSL1 is a versatile transcription factor that binds to GC-rich repressor elements, including the SM MHC promoter GC repressor. Treatment with PDGF-BB, a growth factor involved in vascular remodeling, increases the amounts of RBPJ protein and its ability to bind to the native SM MHC GC repressor. The negative correlation between GC methylation and SM MHC expression is observed in aortic VSMCs. The GC that undergoes methylation acts as a repressor and interacts with RBPJ/CSL-1. This interaction has the potential to inhibit the expression of marker genes associated with VSMC phenotypic modulation in human aortic VSMCs. This finding has important implications for understanding how VSMC-specific and Notch/RBPJ-dependent gene expression is regulated [201].

Bone morphogenetic protein receptor type 2 (BMPR2) is a gene involved in cell development and division [202,203]. The loss of BMPR2 activity has been linked to increased toxicity in PAH [204,205], and aberrant hypermethylation of the BMPR2 promoter region has been reported in familial PAH patients [206], suggesting that DNA methylation may suppress BMPR2 gene expression. Irregular activity of DNMT and MeCPs, proteins that bind to methylated CpG [5′-C-phosphate-G-3′] sites, can disturb DNA methylation and play a role in the development of disease characteristics [207,208]. The SIN3 (switch-independent 3) complex, which includes the SIN3a and SIN3b corepressors, HDAC1/2, MeCP2, and other transcriptional machinery-regulating proteins, has been reported to regulate epigenetic changes [209,210,211]. Although SIN3a does not attach to DNA on its own, it serves as a scaffold for various transcriptional partners and transcription factors with unique DNA-binding capabilities, resulting in the activation or repression of target genes [212]. According to Bisserier M. et al., SIN3a dysregulation in human and animal models is closely related to reduced BMPR2 expression. The upregulation of SIN3a was found to decrease the proliferation of human pulmonary arterial VSMCs while increasing BMPR2 expression by inhibiting the methylation of the BMPR2 promoter region. In a study, it was observed that SIN3a downregulated the expression of DNA and histone methyltransferases, including DNMT1 and EZH2 (enhancer of zest 2 polycomb repressive complex 2). In contrast, it increased the expression of the DNA demethylase TET1. Additionally, SIN3a increased BMPR2 expression by reducing the binding of CTCF, a CCCTC-binding factor, to the BMPR2 promoter [213].

## 4. DNA Methylation in Various Diseases

DNA methylation plays a critical molecular role in the pathomechanisms of various diseases, influencing gene expression and cellular function. Aberrant DNA methylation patterns have been implicated in the development and progression of numerous diseases, including cancer, metabolic disorders, neurodegenerative diseases, and autoimmune diseases [214,215,216].

In cancer, abnormal DNA methylation patterns at imprinted loci are commonly observed across a range of cancer types, including colon, breast, liver, bladder, Wilms, ovarian, esophageal, prostate, and bone cancers. Dysregulated DNA methylation contributes to oncogenesis by either silencing tumor suppressor genes or activating oncogenes. Mutations in DNMTs, differential expression levels of DNMTs, and the dysregulation of TETs are frequently observed in cancer, highlighting the strong association between DNA methylation and cancer [217].

In metabolic disorders, comparing DNA methylation levels in islets between patients with type 2 diabetes and healthy individuals revealed differences in gene expression. Specifically, the decreased methylation of the cyclin-dependent kinase inhibitor 1A (CDKN1A) and phosphodiesterase 7B (PDE7B) promoters impaired glucose-stimulated insulin secretion [218]. Additionally, a shortened leukocyte telomere length was linked to the altered DNA methylation of LINE-1, potentially increasing the risk of type 2 diabetes [219]. In obesity, the increased DNA methylation of hypoxia-inducible factor 3 alpha was found to correlate with higher body mass index, while the DNA methylation of genes like the aryl hydrocarbon receptor repressor was linked to maternal body mass index and birth weight [220]. These findings suggest a complex interplay between DNA methylation patterns and metabolic traits.

Neurodegenerative diseases such as Alzheimer’s disease and Parkinson’s disease exhibit DNA methylation changes in neuronal cells, affecting the expression of genes involved in neuronal function, synaptic plasticity, and neuroinflammation. These epigenetic alterations may exacerbate neurodegeneration and cognitive decline [221].

Autoimmune diseases like rheumatoid arthritis, systemic lupus erythematosus, and multiple sclerosis also feature aberrant DNA methylation patterns that influence immune cell function, cytokine production, and inflammatory responses. Altered DNA methylation in immune cells can dysregulate immune tolerance and promote autoimmune pathology [222].

Overall, elucidating the molecular role of DNA methylation in disease pathomechanisms provides valuable insights into disease etiology, progression, and potential therapeutic targets. Understanding how DNA methylation contributes to disease pathogenesis may pave the way for the development of novel epigenetic-based therapies for treating various diseases.

## 5. Conclusions

VSMCs are an uncommon cell type that exhibits different characteristics during growth and disease. When a blood vessel is injured, VSMCs undergo phenotypic switching, leading to arterial remodeling. This transition is marked by a decline in contractile ability, aberrant growth, migration, and matrix release. Although artificial characteristics are implicated in several cardiovascular diseases, they also play an essential role in vascular injury healing. Recently, key regulators and signaling networks involved in this process have been identified, offering new insights into disease formation and potential therapeutic targets. While important transcriptional regulators of the VSMC phenotype have been identified, DNA methylation has emerged as a crucial player in epigenetic regulation. DNA methylation negatively correlates with gene expression, and it plays a significant role in regulating gene expression.

## Figures and Tables

**Figure 1 ijms-25-03136-f001:**
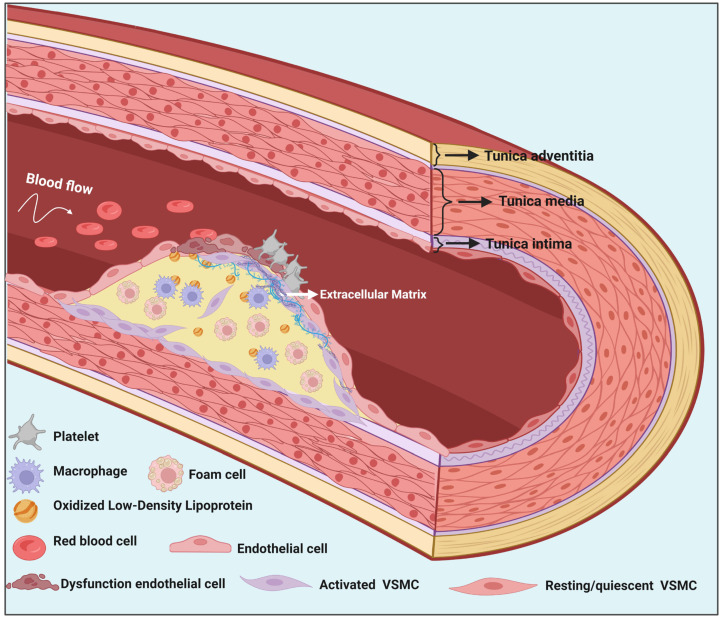
Atherosclerosis: an overview. Atherosclerosis is a complicated and progressive cardiovascular condition marked by the accumulation of fatty deposits, inflammatory cells, cholesterol, and other chemicals inside the walls of arteries. This buildup, known as plaque, can narrow arteries and eventually block blood flow through the arteries, leading to cardiovascular complications. When endothelial cell dysfunction is caused by factors such as high blood pressure, smoking, high cholesterol levels, and oxidative stress, it allows for the infiltration of lipids (such as LDL cholesterol) and immune cells into the arterial wall. Here, lipids, mainly low-density lipoprotein (LDL) cholesterol, penetrate the damaged endothelium and accumulate in the subendothelial space. Immune cells are attracted to the injury site and engulf the accumulated LDL cholesterol, forming lipid-laden foam cells. Smooth muscle cells in the arterial wall begin to proliferate and migrate from the media to the intima of the artery. As smooth muscle cells accumulate, they secrete ECM components, mainly collagen. This forms a fibrous cap over the lipid-rich core of the plaque.

**Figure 2 ijms-25-03136-f002:**
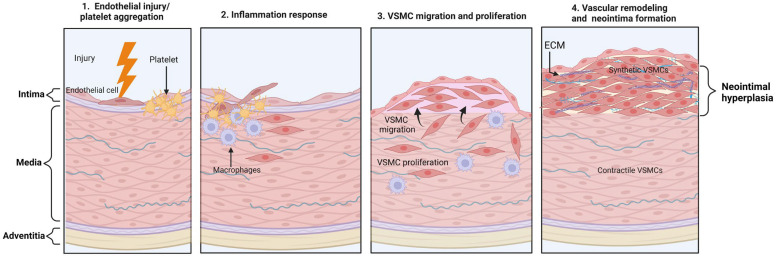
The mechanism of neointimal hyperplasia. Neointimal hyperplasia is a pathological process characterized by the excessive proliferation and migration of smooth muscle cells within the innermost layer of blood vessels, known as the intima. This process often occurs in response to vascular injury, such as after angioplasty or stent placement, and can lead to the narrowing or occlusion of blood vessels, contributing to conditions like restenosis. It is often initiated by some form of vascular injury, leading to VSMC activation and migration. After migrating to the intima, VSMCs begin to proliferate rapidly in response to the growth factors and cytokines, and they also produce excessive amounts of ECM components. Finally, VSMCs accumulate in the intima and continue to secrete ECM. As a result, a thickened layer called the neointimal forms, narrowing the vessel lumen.

**Figure 3 ijms-25-03136-f003:**
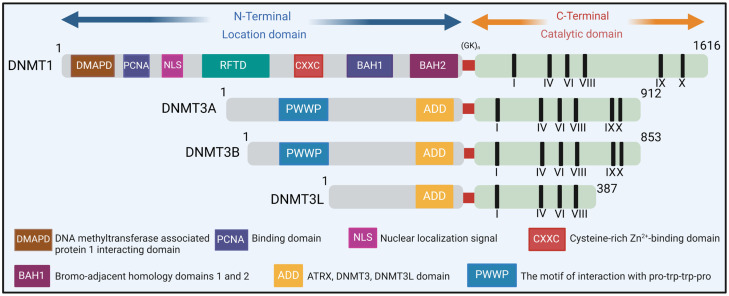
Domain structures of mammalian DNA methyltransferases (DNMTs). DNMT1, the first identified methyltransferase, plays a vital role in maintaining DNA methylation during replication. DNMT3A and DNMT3B are de novo DNA methyltransferases that share similar domain structures [162]. DNMT3L, closely linked to the catalytic region of DNMT3A/3B, lacks catalytic activity. While its N-terminal regulatory domains show little similarity, the catalytic domains of DNMTs remain conserved. (NLS: nuclear localization signal; CXXC: two cysteines separated by two other residues; BAH1/2: tandem bromo-adjacent homology; PWWP: Pro-Trp-Trp-Pro; ADD: ATRX-DNMT3-DNMT3L; (GK)n: glycine lysine repeats).

**Figure 4 ijms-25-03136-f004:**
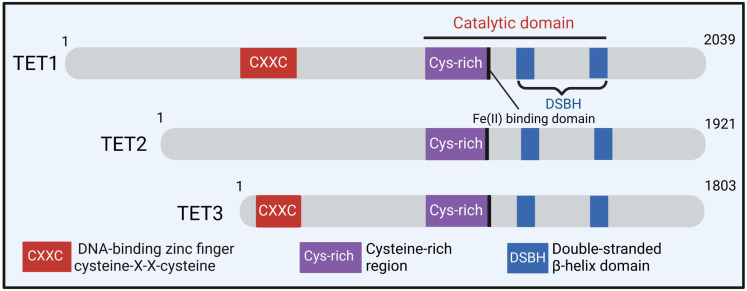
The isoforms of ten-eleven translocation proteins (TETs). TETs, such as TET1, TET2, and TET3, have a catalytic domain (CD) that houses a double-stranded-helix DSBH domain (DSBH) and a domain with a lot of cysteines. The N-terminal of TET1 and TET3, but not TET2, includes a zinc finger domain of the CXXC type. An Fe (II) binding domain is found inside the CD in the C-terminal region of TETs.

**Figure 5 ijms-25-03136-f005:**
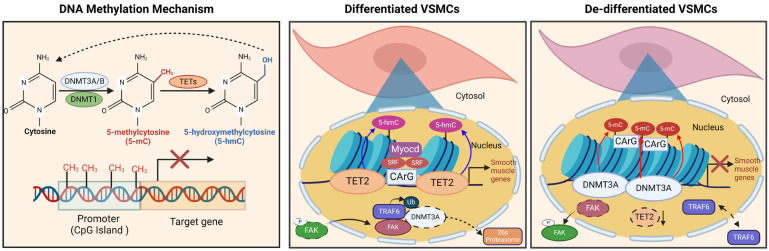
Mechanism of DNA methylation in VSMCs. DNA methylation, a key epigenetic modification, modulates gene expression and cellular identity without altering the DNA sequence. This process involves adding a methyl group (-CH3) to the fifth carbon of cytosine bases, commonly within a cytosine–guanine (CpG) dinucleotide context. DNA methyltransferases (DNMTs) catalyze this modification, while ten-eleven translocation (TET) proteins convert 5-methylcytosine into 5-hydroxymethylcytosine (5hmC). In differentiated VSMCs, low DNA methylation levels, facilitated by destabilizing DNMT3A, help to maintain their differentiated state. Conversely, during pathological conditions, increased DNA methylation contributes to VSMC de-differentiation.

## Data Availability

Not applicable.

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
