# Peer review of "Influence of DNA Methylation on Vascular Smooth Muscle Cell Phenotypic Switching"

_ijms, 2024, doi:10.3390/ijms25063136_

Round 1
Reviewer 1 Report
Comments and Suggestions for Authors
Chanthong Yorn and colleagues review the importance of DNA methylation in regulating the phenotypic plasticity of vascular SMCs and its impact on vascular diseases. The review is concise and well structured. It summarizes the general aspects of SMC phenotypic change and its contribution to different vascular diseases, as well as the most significant findings on DNA methylation. This review would be important for professionals and students who would be introduced to this specific topic. However, there are some points that need attention:
1-The description of diseases due to phenotypic change of VSMC is somewhat repetitive. I would explain the general aspects of VSMC invasion into the intima and then explain in more detail the specificity of each disease without explaining the same invasion each time. Also, single-cell RNA-seq work is not included in the review and I think the audience would benefit from it (for example, in atherosclerosis, the work of the Gary Owens and Helle F. Jørgensen laboratories. For more details, see the Review by Liu y Gómez Arterioscler Thromb Vasc Biol. 2019;39:1715–1723.DOI: 10.1161/ATVBAHA.119.312131).
2-Paragraphs 167-169 state: “Epigenetic modifiers regulate how cells control gene expression and function through mechanisms such as DNA methylation, acetylation, histone modification, and gene-based alterations.” RNA”.
Delete acetylation because it is included in histone modifications. Also, I wouldn't say RNA-based alteration but rather RNA-based regulation.
3- DNA demethylation should be better explained. In paragraph 216 there is a typographical error: the authors state that DNA methylases are not described. It should indicate demethylases.
4- In paragraph 226 they indicate that 5-aza “decreases the expression of the methylation gene.” The authors should explain it in more detail for better understanding.
5- In paragraph 198, the authors state: "First, the methyl group can physically restrict the interaction of DNA with transcription factors such as RNA polymerase, enhancers, and actin." This is not correct. RNA polymerase and transcription factors are not the same. The connector “such us” is maybe a typo? In addition, why the authors mention actin here?
6-In paragraph 251, the denatured matrix is ​​not a concept that the audience will easily understand. It should be explained better.
Comments on the Quality of English LanguageEnglish should be revised
Reviewer 2 Report
Comments and Suggestions for Authors
The manuscript submitted by Yorn et al and entitled “Epigenetic Influence on Vascular Smooth Muscle Cell Phenotypic Switch through DNA Methylation” reviews some aspects of the role of DNA methylation in the clinically highly relevant process of Vascular Smooth Muscle Cell (VSMC) pathophysiology. Indeed, as the authors point out, several VSMC-related pathology exist, which are frequent cause of morbidity and among the most frequent ones of mortality in the Western society. Therefore, understanding the molecular mechanisms behind these processes are highly important as they can reveal future treatment opportunities. Epigenetics is one of these potential mechanisms.
Major comments:
1/ The authors enumerate and describe some frequent and important VSMC-related diseases. Unfortunately, they concentrate mostly on diseases concerning the tunica intima. The authors should summarize the effect of diseases affecting the tunica media, where most of the VSMC reside. In particular, the authors should describe the vascular effect of chronic kidney disease (CKD) and diabetes. In addition, several rare hereditary diseases cause phenotypic switch of the VSMC in the tunica media. The authors should summarize these diseases (e.g. GACI, PXE, Keutel-disease, etc).
2/ In the second part of the review the authors focus mostly on DNA methylation and not on a general epigenetic switch. They should modify the title accordingly.
3/ The authors should include a more comprehensive but not too long description of epigenetics. The current text starting in line 165 is often not sufficiently precise. In addition, the authors should avoid mentioning the role of epigenetics before introducing the concept itself (line 96).
4/ The authors describe mostly specific examples of correlation of DNA methylation and VSMC switch. They should include more examples, which strongly suggest the molecular role of DNA methylation in the pathomechanism of the diseases.
5/ Citations are often not up-to-date enough and a large number (~15% of all references) are more than 20 years old.
Minor comments:
Line 25 “type collagen type” should be corrected
Line 60 “less than” should be corrected to almost
Line 90 “and even mortality” should be corrected to “and can often be lethal”
Line 115 The sentence starting with “VSCMs in IH…” is unclear. What does it mean that VSMC are natural consequences?
Line 177 “generally” is not needed; “cytosine molecules” should be corrected to “cytosine bases in CpG dinucleotide context”. CpG dinucleotides and CpG islands are not the same!
Line 179 “gene and non-coding” should be corrected to “coding and non-coding genes”
Line 185 “coast” should be corrected to “shores and shelves”
Line 187 Unclear text: “DNA methylation heightened” should be corrected.
Line 190 redundant text, see lines 177-181
Line 199 RNA polymerase, enhancers, and actin are not transcription factors, please correct.
Line 201 “proteins like MBD2” rather MeCP2
Line 208 DNMT3A and B have equal affinity to non-methylated and hemymethylated CpGs? This does not seem to be correct. Actually, they participate in maintenance methylation but their primary role is de novo methylation. Please include references.
Fig 3 “Aa1” should be changed to +1
Line 230 The description of 5azaC mechanism is incorrect. It is not specifically targeted to hypermethylated regions. Please correct.
Line 232 The last sentence of the paragraph is unclear. How comes that the inhibition of methylation ultimately restores the methylation of TeT2? Please correct.
Fig 4 Aa1 should be changed to +1; N and C could be omitted.
Line 253 “significant changes DNA methylation” please include “of”.
Line 257 “collage” should be corrected to “collagen”.
Line 288 The paragraph is unclear. Does FAK phosphorylate DNMT3A? How does DNMT3A get ubiquitinated? What is the role of FAK in this process? How does the last sentence explain the role of both de novo methyltransferases when both were downregulated in the cited study? (ref 88)
Figure 5 CaRG should be explained in the legend. FAK should be incorporated in the figure.
Comments on the Quality of English LanguageOnly a few minor typos were observed in the text.
Reviewer 3 Report
Comments and Suggestions for Authors
Dear Authors,
The manuscript entitled "Epigenetic Influence on Vascular Smooth Muscle Cell Phenotypic Switch through DNA Methylation" represents a good study in the field. In my opinion, the whole review has been well conducted by the authors. Only minor comments i have for the submiited manuscript.
1) Please check the whole manuscript for any grammar or phrase error.
2) I'd expected from the authors to add a seperate section of clinical trials with the VSMCs and if they can be utilized for the better administration of human diseases.
These are my only comments. Well done to the authors!!
Reviewer 4 Report
Comments and Suggestions for Authors
The article focuses on assessing the impact of DNA methylation on VSMC phenotype switching. This review does not contribute much new knowledge, as many articles have been published on the relationship between epigenetic modifications and the functioning of VSMCs and their role in diseases related to the improper functioning of VSMCs. However, the journal to which the manuscript was submitted has not recently published a review on this topic. Here are my comments:
Figures describing the structure of DNMT and TET are not necessary in this work. To improve the manuscript, I suggest creating a figure or graphic abstract summarizing changes in DNA methylation regarding the described proteins such as FAK, COL, OPN, and others, highlighting their role in VSMC functioning.
In paragraph 2, line 54, I suggest listing the diseases after a colon and then describing them. In paragraph 3, line 250, the term ECM has already been used and explained earlier, so the abbreviation should be used instead of the term matrix.
In terms of text, a large part is devoted to the description of diseases and the mechanism of methylation. Could the authors try to shorten these fragments and emphasize more how important DNA methylation is for VSMC switching, also in the context of treatment? Please specify what program was used to create the figures.
Round 2
Reviewer 2 Report
Comments and Suggestions for Authors
The authors have considerably improved the manuscript by answering my and the other reviewers’ criticisms.
However, they have only partially answered my first comment. I specifically asked the description of the effect of CKD, diabetes, GACI, and PXE on the VSMC of the tunica media of the arteries. In all these diseases severe arteriosclerosis develops. This has been correctly described for GACI but not for the three other pathologies. Further, the most recent citation for PXE dates from 2014. Since than several scientific breakthroughs were achieved in the field, none of them were cited. The three mentioned diseases require a better description.
Line 273: DNA methylation does not typically occur in CpG islands. Actually, CpG islands (CGI) were initially discovered as non-methylated CG-rich fragments of the genome. This does not preclude the methylation of CGIs but the methylation simply typically occurs at CpG dinucleotides. This should be corrected.
Line 280: First, the methyl group can physically restrict the interaction of DNA with tran- 280 scription factors, RNA polymerase, and enhancers. > Please correct.
Line 283: Instead of MBD2 rather MeCP2 or MBD1.
Author Response
The authors have considerably improved the manuscript by answering my and the other reviewers’ criticisms.
However, they have only partially answered my first comment. I specifically asked the description of the effect of CKD, diabetes, GACI, and PXE on the VSMC of the tunica media of the arteries. In all these diseases severe arteriosclerosis develops. This has been correctly described for GACI but not for the three other pathologies. Further, the most recent citation for PXE dates from 2014. Since than several scientific breakthroughs were achieved in the field, none of them were cited. The three mentioned diseases require a better description.
Response: We appreciate the insightful feedback provided by the reviewer. We have revised the sections addressing the effects of CKD and diabetes, elaborating on their influence on VSMCs (see pages 5-6, lines 183-244). Additionally, we have updated the discussion on PXE, highlighting its implications for the VSMC function (see page 7, lines 265-290).
Line 273: DNA methylation does not typically occur in CpG islands. Actually, CpG islands (CGI) were initially discovered as non-methylated CG-rich fragments of the genome. This does not preclude the methylation of CGIs but the methylation simply typically occurs at CpG dinucleotides. This should be corrected.
Response: We acknowledge the oversight in our previous statement and have rectified it by providing updated information regarding DNA methylation on page 7, lines 298-301.
Line 280: First, the methyl group can physically restrict the interaction of DNA with transcription factors, RNA polymerase, and enhancers. > Please correct.
Response: We have changed the sentences.
Line 283: Instead of MBD2, MeCP2 or MBD1.
Response: We have changed it.
Round 3
Reviewer 2 Report
Comments and Suggestions for Authors
The authors have corrected the manuscript and the text is improved now.